# Management of food socialization for children with Prader-Willi Syndrome: An exploration study in Malaysia

Puspa Melati Wan[1,2]*, Affezah Ali[1,3], Elise Mognard[4,5,6], Anasuya Jegathevi Jegathesan[7], Soon Li Lee[8‡], Rajalakshmi Ganesan[1,3‡], Mohd Ismail Noor[4,5,9], Amandine Rochedy[10‡], Marion Valette[11‡], Maïthé Tauber[11‡], Meow-Keong Thong[12,13], Jean-Pierre Poulain[4,5,10]

1 School of Liberal Arts and Sciences, Taylor's University Lakeside Campus, Subang Jaya, Malaysia, 2 Eradicating Poverty Impact Lab, Taylor's University, Subang Jaya, Selangor, Malaysia, 3 Mental Health and Well-Being Impact Lab, Taylor's University, Subang Jaya, Selangor, Malaysia, 4 School of Food Studies & Gastronomy, Taylor's University Lakeside Campus, Subang Jaya, Malaysia, 5 Centre for Asian Modernisation Studies, Malaysia, 6 Food Security and Nutrition Impact Lab, Taylor's University, Subang Jaya, Selangor, Malaysia, 7 Faculty of Psychology and Social Sciences, University of Cyberjaya, Cyberjaya, Malaysia, 8 Jeffrey Cheah School of Medicine & Health Sciences, Monash University, Subang Jaya, Malaysia, 9 Centre for Community Health Studies, Faculty of Health Sciences, Universiti Kebangsaan Malaysia, Bangi, Malaysia, 10 CERTOP-CNRS, UMR-CNRS 5044, University of Toulouse 2 / ISTHIA, Toulouse, France, 11 Toulouse Children Hospital—Reference Centre for PWS, INSERM U1043, University of Toulouse 3, Toulouse, France, 12 Genetic Medicine Unit, University of Malaya Medical Centre / Faculty of Medicine, Universiti Malaya, Kuala Lumpur, Malaysia, 13 M. Kandiah Faculty of Medicine and Health Sciences, Universiti Tunku Abdul Rahman, Kampar, Malaysia

☙ These authors contributed equally to this work.
‡ LSL, RG, AR, MV and MT also contributed equally to this work.
* wanpuspamelati@gmail.com

**Data Availability Statement:** All relevant data are within the manuscript. The data utilized in this study were sourced from a combination of observation and reflective interviews. A sample of

## Abstract

This study aims to explore the food management strategies among caregivers/family members of children with Prader–Willi Syndrome (PWS) using the lens of 'familialisation' of a health problem and the sociology of food socialization. Food intake among individuals with PWS is a main concern for parents, caregivers, and medical practitioners as it affects their physical, mental, and social well-being throughout their lives. Earlier studies on PWS and food intake centered around dietary management, dietary intake and growth, nutritional treatment and pharmacological approaches, nutritional phases, and weight gain. However, little has been done to understand the challenges of managing children with PWS from the sociological lens of food management strategies and socialization among families in Malaysia. This study is based on an investigation involving eight children with PWS and 46 family members and caregivers through lab observations and reflexive interviews. Ten food management strategies were identified that were adopted by the caregivers and families, which were influenced by cultural factors, family norms, and formal and informal support systems. The findings will influence future behavioral interventions to ensure the empowerment and well-being of individuals with PWS and their families.

interview transcript and output data which also include highlighted observed data with a participant from the study is available for reference, offering insights into the data used in this study. However, in adherence to confidentiality as requested by the participants, access to the observational recordings cannot be provided directly. Interested parties are kindly directed to contact the lead researcher at wanpuspamelati@gmail.com.

**Funding:** This study is a part of the Hubert Curien Partnership France-Malaysia Hibiscus (PHC Hibiscus) Grant titled "The Socialization of eating practices in children with Prader-Willi syndrome" (MYPAIR/1/2020/SS05/TAYLOR/1), funded by Ministry of Higher Education (MOHE), Ministry of Europe and Foreign Affairs (MEAE) and Ministry of National Education, Higher Education and Research (MESRI), France, which is a mirror study of "Socialisation des Pratiques alimentaires des Enfants avec un Syndrôme Prader-Willi" (SoPAP – translation Socialisation of Food Practices of Children with Prader-Willi Syndrome). The URL: https://www.moe.gov.my/en/pemberitahuan/media-statement/deklarasi-bersama-mengenai-pelancaran-rasmi-perkongsian-penyelidikan-hubert-curien-malaysia-perancis. The PLOS cost of publication was supported by the chair of Food Studies of Taylor's Toulouse University Center (TTUC). The funders play an important role in supporting the research direction and publication requirement. In the capacity of providing financial aid, the funders had no role in study design, data collection and analysis, decision to publish, or preparation of the manuscript.

**Competing interests:** The authors have declared that no competing interests exist.

## Introduction

As a complex neurodevelopmental disorder, Prader–Willi syndrome (PWS) represents a developmental challenge for children, especially regarding food. Recent examination of the natural history of PWS suggests a more complex progression leading to four main nutritional phases, with subphases for the first two phases [1]. Not all individuals with PWS experience all the phases and subphases, and their progression may be further altered using growth hormones [2]. Medical experts agree that behavioral interventions are possible, although they are yet to be well defined. In the early developmental phase, interventions to ensure proper nutrition include the utilization of special teats [3]. However, drastic measures, such as tube feeding [4, 5] and swallowing training, are common among infants with PWS [4–6].

Other caregiver-suggested interventions include restrictive feeding and strict meal habits [7], monitoring of low-calorie and well-balanced diets, portion control, personalized nutrition [8] and ensuring regular physical exercises for children with PWS. Miller and Tan [9], suggested that mealtimes should be less stressful for the family. To decrease behaviors centered around food for the individual with PWS, recommendations were made for the whole family to eat the same, well-balanced, healthy diet and drink plain water or milk with meals. They found that these recommendations resulted in improved weight control for the individual with PWS and more peaceful meals for the family, without arguments regarding differences in diet between family members.

Another common method of intervention is appropriate psychological and behavioral counseling of individuals with PWS and their caregivers [10] to manage symptoms and behavioral problems faced by PWS adolescent, including food theft, overeating, and tantrums. Interventions such as the reward system has been found by some to improve social skills and provide relevant tools for anger management. Another approach was family therapy, through which the counselor worked with the parents to establish a consistent parenting style and to standardize the family menu and structured daily routines [11].

Food habit concern among individuals with PWS is exacerbated in certain countries, such as Malaysia. First, as demonstrated by the Malaysian Food Barometer, food habits are characterized by a high rate of eating food prepared outside the home [12–16]. Second, the household structures commonly encompass several generations under the same roof and around the family dinner table. It is also common in Malaysia to have extended family members living as immediate neighbors which the children visit frequently.

The available literature emphasizes on culture of individual autonomy and choice, even in food habits, while Asian societies tend to value intergenerational exchange and family responsibility [17]. Therefore, there are important social expectations and practices regarding the involvement of grandparents in raising grandchildren [18], although the norms and expectations may vary across the culturally diverse population of Malaysia. Finally, live-in domestic helpers are present in many households in Malaysia. There were 130,450 registered domestic helpers in Malaysia in 2018 [19]. Although the number was reported to have decreased during COVID-19 pandemic, those still serve as domestic helpers may play an important role in managing food habits among children with PWS.

This research adopts a point of view that articulates two complementary perspectives to study the modality of "problematic" food behavior among children with PWS and its consequences on their family lives. These two perspectives are the "familialisation" of a health problem and the sociology of food socialization. The concept of familialisation is used to analyze the way a family understands and reinterprets the messages provided by medical experts during and after the diagnosis. These messages concern the disease, its cause, the modalities of its evolution, and how to care for a person with such a disease. The focus extends to the family roles to manage the presence of a person with health problems in everyday life.

The concept of familialisation appeared first in the frame of medicalization studies. The notion of "de-familialisation" was used to analyze the transfer of activities and responsibilities from the frame of the family to state or private care systems [20]. This conceptual frame allowed to develop international comparisons [21–23] and to study the implications on gender equality [24]. An alternative formulation of the concept of familialisation have been used by some French scholars such as Keppens (2010), Martin et al. (2016), and Rochedy et al. (2023) to describe the way family members understand, reinterpret and finally use the advice of medical experts to re-organize family life, including the redistribution of parental roles [25–27]. This concept has been further articulated in the Food Social Norms Internalization (FSNI) theory based on a study of families with PWS child [27]. This process relates to how the family of an individual with PWS appropriates messages and advice formulated by health experts at the time of diagnosis. Exploring these concerns would provide important insights towards intervention initiative and case management [28].

Familialisation concerns several questions: How do parents understand the explanation given by the medical experts on the syndrome-specific trajectory that goes from anorexic behavior at the beginning of the life to hyperphagia that lasts from childhood to adulthood and is characterized by strong cravings for food and relentless thinking about it? How in such a context does the family manage the supervision of food practices? Which educative strategies do they consider? To what extent are the employed strategies informed by knowledge and advice from the healthcare team? Who are the actors involved in the caregiving processes, and who tends to take the responsibility? How many significant others are involved in the supervision of food practices, and what social position do they hold in the family (i.e., mother, grandmother, domestic helper, uncle, grandfather)?.

These research questions connect with the sociology of food regarding the internalization of rules and social norms [29, 30]. The internalization of food behaviors not only allows a child to eat in society as they learn about table manners, such as the use of tools, body control, postures, chewing habits, body noises but also food gratification, and identification of common and individual spaces at the table. Some authors have suggested that during childhood, individuals typically go through a so-called "neophobia" phase [31–33]. Throughout the neophobic cycle, the social interactions between parents, caregivers and the child will contribute to the internalization of the food norms [34]. The process of food socialization among children with PWS is further exacerbated as their condition is often tied to concerns surrounding food control [35]. While most of the medical literature dedicated to autism clearly identifies the issues related to food habits, they have been mainly problematized in terms of nutritional risks. A few works have proposed a problematization from the social sciences for autism at large [36, 37] or more specifically the subjects with PWS [27, 38].

PWS in Malaysia has evolved from being supported under the umbrella of rare diseases to a disease that has specific support groups and interventions. Medical and healthcare providers since 80s recognized that PWS management was not about managing the child's medical needs but requiring the whole family to understand and support initiatives. Early support came from family activities and family camps, organized by government hospitals and clinic staff. These camps reduced stigma related to PWS, built relationships between families, and provided social and mental support. Interventions expanded beyond medical advice and focused on familial interactions, home nursing procedures, and social adjustments. In 2004, families collaborated to establish the Malaysian Rare Disorders Society, a non-profit organization dedicated to providing support, education, and empowerment to families. In 2016, the Persatuan Sindrom Prader-Willi Malaysia (PWSAM) was formed to foster awareness, support individuals and families, and provide accurate information to the public. Scholarly investigations into PWS in Malaysia are lacking [39], with researchers focusing on the medical aspects

of an ailment, namely on prevalence and treatment strategies [40–42] to general chromosomal abnormalities and on disabilities in general. Studies investigating the symptom management of PWS in children in Malaysia, especially their insatiability for food and family interactions, are yet to be conducted [43]. Malaysian caregivers struggle to care for their children with limiting conditions due to unavailable or inaccessible information [44] as well as facilities and care services [45]. Therefore, this study aims to investigate the process of (de)familializing food management in families with PWS children and understand the food socialization of PWS children in the Malaysian context.

## Materials and methods

The data obtained from this research is part of a larger research cooperative set-up for the Hubert Curien Partnership France-Malaysia (Hibiscus) research titled "The Socialization of eating practices in children with Prader-Willi syndrome" (MYPAIR/1/2020/SS05/TAYLOR/1), which aims to compare findings from France and Malaysia. The cooperative takes place in the research activities of the Centre Prader Willi France at Toulouse University Hospital, directed by Professor Maïthé Tauber in partnership with CERTOP-CNRS 5044 of the University of Toulouse and ISTHIA. This Hibiscus project is a partnership between Toulouse University in France and Taylor's University in Malaysia within the framework of the Chair "Food Cultures & Health." In France, it takes place at the OVALIE experimental platform of Toulouse University and in Malaysia at the Social Behavioural Lab situated at Taylor's University.

The methodology are three complementary techniques: 1) expert interviews, including of members of a medical team as well as founders of national associations, 2) observations of family meals and 3) reflexive interviews with individuals with PWS and their family members. The observation is followed by a reflexive interview during which extracts of the recorded meal sessions were shown and used to support the discussion. The interviews provided macro-level context for the available support in Malaysia, while the observations gave micro-level insights into the interactions among family members and the strategies, they use to manage the eating behaviors of children with PWS.

For the Malaysian research project, the research procedure was approved by the Ethics committee of Taylor's University (Reference No: HEC 2020/149) and the University of Malaya Medical Centre (UMMC) (MREC ID No: 2021112–9711). The recruitment was done between January–March 2022 with the support of PWSAM and UMMC. All participants and their legal representatives provided written informed consent before participating in the study on the research procedures and rights as well as for their information to be published. With a focus on the strategies used by families to manage the eating behavior of their children with PWS and uncovering the transformations and development of their children over time, this article draws from the findings of the above methods.

Following the research briefing, the session for lab observation with the families was arranged at the Social Behavioural Lab at Taylor's University. Lab observation is an important feature of this study to enrich the understanding of social interaction on eating habits [46–48]. The families were allowed to invite any family members who frequently ate together with the individual with PWS. As a result, the number of participants ranged from four to ten persons per session, with a total of 53 participants (refer to Table 1 below). The room was set up with a long 10-seat table in the middle of the room with chairs, including highchairs, and the family decided on their seating arrangements. The food was placed on a side table, and the family was instructed to serve themselves as they usually would at home as part of the observation.

The reflexive interviews were conducted at the Social Behavioural Laboratory at Taylor's University as a continuation of the meal observations. The reflexive interviews followed three

**Table 1. Summary of participants of the meal sessions at Taylor's Behavioural Lab.**

| Family Lunch Size (Number of persons attended) | Relationship to PWS Participants | Age of PWS Participants (years) |
|---|---|---|
| 5 | Father, Mother, Sister | 21 |
| 7 | Father, Mother, Brother, Sister | 10 |
| 7 | Mother, Brother, Sister Domestic Helper | 12 |
| 8 | Father, Mother, Auntie, Uncle, Cousin | 6 |
| 4 | Father, Mother, Sister | 24 |
| 10 | Father, Mother, Brother, Sister, Grandmother, Grandfather, Auntie, Cousin | 3 |
| 7 | Father, Mother, Brother, Sister, Grandmother, Domestic Helper | 2 |
| 5 | Father, Mother, Brother, Sister | 12 |
| 53 persons | | |

[a] Number of persons in attendance, including individuals with PWS

main phases. During the first part of the interview, families were asked to reflect on excerpts from the video recordings of the meal, which were recorded and edited during the session at the Social Behavioural Lab. Among the selected excerpts were social interaction difficulties, table rules, and taste-related emotions, with the objective of encouraging the families to discuss their practices. The first part of the interview aimed at strengthening credibility, where the researchers assessed participants to ensure that their interpretations were credible and in line with the participants' comprehension of their lived experiences. The second part of the interview focused on the neophobia of the child with PWS and its transformation over the years. This phase was supported by sharing previous findings from the French research team [33, 35–37], which aimed to guide the participants to extract themselves from the current meal observation and adopt the position of "experts." The third and final stage focused on the strategies developed by the family to manage the eating behavior of their child with PWS and the transformation of these strategies throughout the development of the child.

The interviews ranged between 30 and 60 minutes in duration and were audio and video recorded. The content of the interviews was transcribed, and a phenomenological data analysis method was used to support the findings of the conversations recorded during the interviews. Significant statements that were highlighted by any family members or children with PWS were gathered from the group interviews and were categorized into meaningful themes pertaining to subjective connotations and interpretations of eating habits, changes, and the socio-culturally embedded nature of family experiences. Significant statements were identified through a process of iterative coding and categorization of the transcribed interviews and observations. The significance of these remarks was determined by their ability to offer perspectives on the real-life experiences of the family members and children with PWS, and their contribution. The identification of significant statements was not exclusively determined by their frequency, but rather by their pertinence and depth in addressing the research objectives.

The identification and development of themes were achieved through a collaborative process that engaged by multiple investigators. At first, each investigator analyzes the transcript separately to uncover initial themes that arose from the data. Following this, the researchers gathered to compare and integrate their respective coding techniques to reach a consensus on the most significant themes that encapsulated the fundamental nature of the data. This collaborative approach ensured the reliability and validity of the thematic analysis and minimizing

researcher biasesness. Moreover, it is crucial to acknowledge that the execution of the theme analysis procedure was done by a collective of researchers possessing varied proficiencies in the fields of sociology, psychology, and public health. The incorporation of several fields of study enhanced the analysis by enabling detailed interpretations of the data from different viewpoints and theoretical frameworks. To ensure reflexivity and rigor in the interpretation of the findings, regular debriefing sessions and regular discussions were performed throughout the analytic process.

## Results

The data were derived from eight families who participated in the study. There were one Indian, two Chinese, and five Malay families with children with PWS aged between 2 and 24 years. The number of participants in each session ranged from four to 10. In all, 53 individuals participated in the meals organized at the Social Behavioural Lab at Taylor's University. Two families brought their domestic helpers, while extended family members included the aunts, uncles, grandparents, and cousins of the children with PWS.

In terms food management strategies and socialization among families in Malaysia, the data revealed that the families tend to rely on experts as soon as their child was diagnosed with PWS, though at a varying degree. It is not uncommon for the families in this study to seek advice from pediatricians, dieticians, geneticists, PWS support communities, among others, when they first learned about the rare condition of their child. However, the data also showed that the families do curate their own management strategies taking into social and cultural context, innovating the professional recommendations given. Below are two example that illustrate the process of familialisation among the interviewed respondents. Strategies employed by the families are indicated in brackets within the case studies, with further elaboration and categorisation provided in Table 2.

### Case 1

For this family, the PWS child was diagnosed at the age of 4 to 5 years. Initially, they had little knowledge about PWS and relied heavily on medical professionals for guidance and information. This marked the start of their familialisation process. They diligently followed the recommendations of medical professionals, including hormone therapy and insulin treatment for the child's diabetes, as advised by the endocrinologist. They also adhered to dietary advice provided by the dietitian regarding the quality and quantity of food intake. Regarding medication and treatment, the family strictly followed the professional recommendations due to their limited knowledge. However, to manage the dietary advice, they took their own initiatives and made adjustments based on their family dynamics and environment. To help stabilize the child's sugar levels, they encouraged exercises such as jogging and stopped purchasing food from outside. The mother took the initiative to cook food at home, with a focus on steamed dishes. The medical professionals advised the family to control the child's daily food intake. As a measure of control, they established a fixed eating schedule with designated times for meals, such as morning, afternoon, and night, with lunch consistently set at 1 pm (Scheduling). The mother took responsibility for serving the child's meals to control portion sizes because when he served himself, he tended to take larger portions. She estimated the portion using the same bowl every day (Bowl system). If the child still felt hungry after finishing the meal, the mother would offer vegetables as a healthier option for additional food (Fruits and vegetables system). After eating, the mother would encourage the child to leave the table and go to another area to avoid seeing or being tempted by other food (Distraction). Additionally, she promptly cleared the table after everyone finished eating to minimize further temptations (Cleaning-up system).

**Table 2. Descriptions of food management strategies.**

| Category | Description |
|---|---|
| Fruits and vegetables system | Family members/caregivers include a lot of fruits and vegetables as a dietary option. Fruits and vegetables were used as the main meals, as water replacement, and as a "healthier" food snack. Additional fruits and vegetables were given during mealtimes and between mealtimes. |
| Instill self-regulation | Family members/caregivers educate their PWS child on their condition and/or food consumption, including what can be eaten and what should be avoided. This includes show-and-tell and label reading. |
| Little bits system | Family members/caregivers allow the PWS child to have a little bit of anything that they ask for, including fried food, high-calorie foods, and sweet food or drinks. |
| Distraction | Family members/caregivers use entertainment via gadgets, television, through play/ activity, or moving to another space to distract the child from food. |
| Scheduling | Family members/caregivers follow a meal schedule to regulate their eating behavior: breakfast, lunch, teatime, and dinner. |
| Bowl system | Family members/caregivers use a special bowl to measure the amount of food to be given to the child with PWS per serving. |
| Food for all | There were two types of "food for all" strategies:<br>Family members/caregivers adopt similar eating habits to the child with PWS. This includes regular meals where all will eat PWS-child-friendly food and no snacking habits or no junk food available at home for all.<br>The child with PWS eats the same food as other members of the family, that is, special dietary restrictions or menu for the child with PWS. |
| Cleaning-up system | Family members/caregivers complete their meals earlier than the child with PWS and quickly clean the table before the child with PWS finishes their food. |
| Dummy system | Family members/caregivers deceive the child with PWS into thinking that they are consuming something else (i.e., feed the child a spoonful of PWS-friendly food before feeding themselves a spoonful of different food or adding food coloring to water to make them think it is a juice). |
| Cheat day | Family members/caregivers allow the child with PWS to eat anything and any amount of food on selected days. |

This initiative indicated that the process of de-familialisation was ongoing and evolving within the Family 1, as they grappled with the complex behavioral challenges associated with PWS and sought guidance from medical professionals. However, the child's tantrums remained a challenge to manage, despite the family discussing the issue with medical professionals. The family highlighted the difficulty they faced in controlling these outbursts.

## Case 2

The child with PWS for Family #5 was diagnosed very early at two months of age. Their familialisation process begins when they actively sought knowledge about PWS from various sources such as local and international medical professionals, dieticians, and PWS support communities. By actively seeking information, they consider themselves prepared to manage their child. The mother has chosen to become a full-time housewife in order to focus on caring for the child with PWS. While assimilating information from professionals, they notice that their child does not exhibit the food-seeking behavior typically seen in other children with PWS. They consulted a medical expert from the UK and discovered that their child is indeed different from others. They believe this may be due to the early behavior management they have implemented since the child was young. From the time the child started chewing and swallowing, the mother consistently reminded her that she is a child with PWS and educated her about the syndrome and the consequences of overeating. Over the years, the child has developed an understanding of what is good and bad for herself, demonstrating self-regulation. Whenever she accompanies her parents to the supermarket, she selects her own food and

Table 3. Matrix of Food Management Strategy (FMA) adoption.

| Food Management Strategies | Family 1 | Family 2 | Family 3 | Family 4 | Family 5 | Family 6 | Family 7 | Family 8 | Total Family Adopting FMS |
|---|---|---|---|---|---|---|---|---|---|
| Fruits and vegetables system | √ | √ | √ | | √ | √ | √ | √ | 7 |
| Instill self-regulation | | | | | | X | √ | X | 3 |
| Little bits system | X | √ | X | X | X | | | | 5 |
| Distraction | √ | | √ | √ | | | X | | 4 |
| Scheduling | X | | X | | | X | X | X | 5 |
| Bowl system | O | √ | X | O | √ | O | | √ | 7 |
| Food for all | | | | | O | X | √ | O | 4 |
| Cleaning-up system | O | √ | O | | O | O | O | O | 7 |
| Dummy system | X | | | | | | | | 1 |
| Cheat day | | X | | X | X | | X | | 4 |
| **Total Strategies by family** | 7 | 5 | 6 | 4 | 6 | 6 | 7 | 6 | |

O–Data recorded based on lab observation (11)

X–Data recorded based on reflexive interview (19)

√ –Data recorded based on both lab observation and the reflexive interview (17)

can identify low-sugar and low-calorie options for herself. She also displays a lack of eagerness when seeing food and patiently waits for her own food to be served, even if it is after everyone else during laboratory observations. She maintains a calm demeanor while eating and does not seek out other people's food. Initially, the family tried preparing healthy meals specifically for her, but eventually realized that their child can eat the same food as everyone else at home as long as they control the portion sizes (Food for all). They do not count calories, as suggested by medical professionals, because they believe that portion control is sufficient for maintaining control. They use the same plate every day as a means of estimating portion sizes (Bowl system). They start with a small portion and gradually add more if she wants it, stopping when it exceeds the plate's capacity. Additionally, they have encouraged her to eat vegetables from young age, emphasizing their health benefits. As a result, she often consumes more vegetables compared to other foods (Fruits and vegetables system). The family is confident in the quality of the food they provide because the mother cooks for the entire family. Although medical professionals have warned about the risk of choking in individuals with PWS, the family has not experienced this issue and therefore pays less attention to food size, focusing instead on ensuring the child chews her food properly. This family has demonstrated de-familialisation process by adaptation to their child's condition, who does not exhibit typical food-seeking behavior, and they have a strong belief in their ability to handle it on their own familialisation.

From the familialisation process identified for each family, 10 main food management strategies curated were identified and described in Table 2: 1) fruits and vegetables system, 2) instill self-regulation, 3) "little bits" system, 4) distraction, 5) scheduling, 6) bowl system, 7) food for all, 8) cleaning-up system, 9) dummy system, and 10) cheat day.

The findings reflect some commonly employed strategies, such as the bowl system, self-regulation [49], and food for all [9]. However, it is interesting to note the other categories appeared to be salient among the families as shown in Table 3.

About 23.40% of the strategies employed by the families were identified during the observation, 40.43% were identified from the reflective interview, and 36.17% were identified from both observation and reflective interview. Most of the families tend to employ the bowl, fruits and vegetables as well as the cleaning-up system (88%). Furthermore, at least 50% of the families engaged in the little bits system, distraction, scheduling, food for all and cheat day. Only

three families reported that they had consciously put in a lot of effort into instilling self-regulation by reading food labels and discussing food consumption practices. Although the other families confessed that they had been introduced to calorie counts or to engaging the children to be more involved in their food selection by the experts, most did not follow through. In terms of the number of strategies adopted by the families, 90% of the families practiced at least five different strategies to manage the food intake. It is interesting to note that three families employed a similar combination of the fruits and vegetables system, little bits system, bowl system, and cleaning-up system.

## Discussion

Adopting the lens of familialisation, many of the strategies used by parents to manage food habits are based on expert advice. Some families were frequent clients of PWS clinic organized by University of Malaya Medical Centre (UMMC) to support families with PWS children. Through these visits, the family members learned not only about the development of their children and their current health status from the pediatricians and geneticists, but they also met the physiotherapists and nutritionists/dietitians who would provide important information to enable the families to rethink their food management. The advice not only shaped the perspectives of the immediate family members; but also influenced informal conversations with other parents including international social support groups.

The support provided by the PWSAM was highlighted by some of these families as an important source of information. Because most of them are active members of the association, they reported that their involvement allows them to keep abreast with the latest happenings around the world through writing and sharing through the association. The family members shared how they learned not only how to manage similar cases, but also about other potential behaviors that their child may exhibit in the future according to their developmental stage. This information enables them to further strategize the food management behaviors of their children with PWS in addition to the entire family and caregivers.

The strategies identified reflect the learning of rules and internalization of food behavior among individuals with PWS. The strategies allow them to learn about acceptable table manners (distraction), food intake behaviors (instill self-regulation, fruits and vegetables system, bowl system), food gratification (cheat day, little bits system), and feeding dynamics when eating with others (i.e., the feeding rotation among family members and domestic helpers, cleaning-up system, food for all).

The data findings reflect learning beyond the table norms and household for these families. Referring to Malaysian culture, some families reasoned that it is important that they allowed their children to be treated as "normal" and not deprive them of local delicacies. Although they understand the importance of "controlling" the food [7], hence the bowl system, cleaning system, and scheduling [46], they also tolerate cakes, curry, *masak lemak*, and ice-cream as part of the little bits system or a cheat day. These strategies may not be ideal according to the recommendations of scholars [7, 9] or medical practitioners; however, these strategies were reported to have been effective for these families to help them avoid meltdowns or self-guilt.

Other cultural factors that influenced food management strategies include the understanding of food culture when visiting others or as practiced by their extended family members. Because Malaysia is known for its hospitality regarding food for guests and family members, it is hard for some caregivers to cut off children with PWS from certain foods completely. Five families mentioned their allowance for the little bits system or providing similar food for the child with PWS (i.e., food for all), especially when they eat with guests or extended family members. This is even more challenging for families with extended families involved in daily

caregiving and preparing meals for these children, which is common among Asians [17] and was evident among the observed respondents. They believed these strategies would allow them to maneuver their cultural consciousness without excluding the child with PWS and uphold their cultural practices.

The cultural aspect extends beyond daily events to special occasions and festivities, such as birthday parties, Eid celebrations, Chinese New Year, and Deepavali. Although one family reported limiting such exposure in the earlier years for their child with PWS, others partook in these events alongside their child with PWS throughout their development. Most of the family members reported that they "briefed" the hosts/family members on the condition of their child and their diet restrictions as a mechanism to manage their food intake. A few parents declared that they adopted a strict parenting approach regarding food management to ensure the well-being of their child and to keep them safe. However, several families opted for the cheat day management strategy for these special days. They acknowledged the cultural norms and believed that they should lax their control in these "once in a while" events. One of the respondents reported the pressure to enable such eating practices during family events due to comments from their elderly relatives and in-laws suggesting that they allow the child to enjoy such events.

Though these findings provide useful insights into the familialisation of food management and food socialization, it is imperative to also note the limitation of this research. First, since the participants were only recruited through PWSAM and UMMC, they may not be representative of the entire population of study. Second, the interview approach and meal observations, though well designed and done thoroughly, may not cover all PWS families' experiences and concerns. And third, monitoring meals in a behavioral lab may not accurately reflect home mealtime dynamics and hence may affect the observation data. Despite these limitations, this research provides a fundamental understanding on the complex relationship between familial dynamics, cultural influences, and eating behavior control in Malaysian children with PWS.

## Conclusion

The findings presented in this study showcase the process of familialisation by families who (re)gained control from medical care experts to (re)arrange family life, internalization of food norms, and (re)organize family roles. The data obtained in this study should be considered in the process of developing behavioral interventions and remedial measures. It is evident that managing the food intake of a child with PWS is complex as the concern extends beyond the selection of food or the determination of the caregivers to control food intake. The intervention should consider the family norms, the roles of significant others, exposure to information, and interpretation of cultural expectations and social experiences. Creating a system that is recognized by all members reduces stress and tension between caregivers/family members and their child with PWS. The internalized strategies empower individuals with PWS to operationalize their food habits and gain control over their lives. This in turn enhances the quality of life of both the caregivers/family members and the individuals with PWS.

## Acknowledgments

The authors would also like to acknowledge the Prader–Willi Syndrome Association Malaysia (PWSAM) and University of Malaya Medical Centre (UMMC) for their assistance in the respondent recruitment of this project. Special thanks to the President of PWSAM, Mr. Azhar Talib, and the Founder of the Malaysian Rare Disorders Society, Dato' Hatijah binti Ayob, whose expert knowledge contributed to providing the necessary context of PWS in Malaysia. Furthermore, we would like to thank Mr. Hafiz Abdullah, who played a vital role in managing

the lab recording and editing, and Tengku Puteri Balqis Tengku Zainal Abidin, the research assistant for this project.

## Author Contributions

**Conceptualization:** Puspa Melati Wan, Affezah Ali, Elise Mognard, Anasuya Jegathevi Jegathesan, Amandine Rochedy, Marion Valette, Maïthé Tauber, Meow-Keong Thong, Jean-Pierre Poulain.

**Data curation:** Puspa Melati Wan, Affezah Ali.

**Formal analysis:** Puspa Melati Wan, Affezah Ali.

**Funding acquisition:** Elise Mognard, Anasuya Jegathevi Jegathesan, Amandine Rochedy, Maïthé Tauber, Meow-Keong Thong, Jean-Pierre Poulain.

**Investigation:** Puspa Melati Wan, Affezah Ali, Elise Mognard, Mohd Ismail Noor, Meow-Keong Thong, Jean-Pierre Poulain.

**Methodology:** Puspa Melati Wan, Affezah Ali, Elise Mognard, Anasuya Jegathevi Jegathesan, Maïthé Tauber, Meow-Keong Thong, Jean-Pierre Poulain.

**Project administration:** Puspa Melati Wan, Affezah Ali.

**Resources:** Puspa Melati Wan, Affezah Ali, Jean-Pierre Poulain.

**Software:** Affezah Ali.

**Supervision:** Puspa Melati Wan.

**Validation:** Puspa Melati Wan.

**Writing – original draft:** Puspa Melati Wan, Affezah Ali, Elise Mognard, Anasuya Jegathevi Jegathesan, Soon Li Lee, Rajalakshmi Ganesan, Mohd Ismail Noor, Amandine Rochedy, Maïthé Tauber.

**Writing – review & editing:** Puspa Melati Wan, Soon Li Lee, Rajalakshmi Ganesan, Mohd Ismail Noor, Amandine Rochedy, Marion Valette, Maïthé Tauber, Meow-Keong Thong, Jean-Pierre Poulain.

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
