## [Decision Letter · Decision Letter 0]

4 Mar 2024

PONE-D-23-43273Management of food socialization for children with Prader-Willi Syndrome: An exploration study in MalaysiaPLOS ONE

Dear Dr. Wan,

Thank you for submitting your manuscript to PLOS ONE. After careful consideration, we feel that it has merit but does not fully meet PLOS ONE’s publication criteria as it currently stands. Therefore, we invite you to submit a revised version of the manuscript that addresses the points raised during the review process.

Dear authors,

You have to incorporate the suggestions of the reviewers before considering for further publication processes. 

With regards,

Ranjit

We look forward to receiving your revised manuscript.

Kind regards,

Ranjit Kumar Dehury

Academic Editor

PLOS ONE

Journal Requirements:

This study is a part of the Hubert Curien Partnership France-Malaysia Hibiscus (PHC Hibiscus) Grant titled “The Socialization of eating practices in children with Prader-Willi syndrome” (MYPAIR/1/2020/SS05/TAYLOR/1), funded by Ministry of Higher Education (MOHE), Ministry of Europe and Foreign Affairs (MEAE) and Ministry of National Education, Higher Education and Research (MESRI),  France, which is a mirror study of “Socialisation des Pratiques alimentaires des Enfants avec un Syndrôme Prader-Willi” (SoPAP – translation Socialisation of Food Practices of Children with Prader-Willi Syndrome). The  URL: https://www.moe.gov.my/en/pemberitahuan/media-statement/deklarasi-bersama-mengenai-pelancaran-rasmi-perkongsian-penyelidikan-hubert-curien-malaysia-perancis. The funder plays an important role in supporting the research direction and publication requirement.

Additional Editor Comments:

Dear authors,

You have to incorporate the suggestions of the reviewers before considering for further publication processes.

With regards,

Ranjit

Reviewers' comments:

Reviewer's Responses to Questions

**Comments to the Author**

1. Is the manuscript technically sound, and do the data support the conclusions?

Reviewer #1: Yes

Reviewer #2: Yes

2. Has the statistical analysis been performed appropriately and rigorously? 

Reviewer #1: N/A

Reviewer #2: N/A

3. Have the authors made all data underlying the findings in their manuscript fully available?

Reviewer #1: No

Reviewer #2: Yes

4. Is the manuscript presented in an intelligible fashion and written in standard English?

Reviewer #1: Yes

Reviewer #2: Yes

5. Review Comments to the Author

Reviewer #1: Wan and colleagues seek to better understand the strategies used by families of those with Prader-Willi syndrome (PWS) to manage the disorder. They report on food socialization and cultural aspects in Malaysia that have important impacts on managing food intake and behavior in individuals with PWS. Drawing from interviews and observation of eight families, the authors identify common food management strategies used by families in everyday life.

Critique:

This is a generally well-written manuscript that explores a relatively under-appreciated aspect in the management of PWS – how families, in the context of their culture, apply the information and guidelines they are given from medical professionals, and adapt it to their individual families. It is important to understand the real-world strategies that families use to manage PWS, which may or may not be fully aligned with the guidance they receive from health care professionals.

Specific comments:

The Introduction is long and should be streamlined; for example, while the description of familialisation and discussion of Malaysian culture and aspects of food/living arrangements are important for understanding the context of the paper, details on the stages of food consumption in children (paragraph beginning at line 138) is not essential and can be shortened or eliminated, and the discussion of the history of PWS support in the country could be much more concise.

More detail should be provided on the methods used for thematic analysis – including, how 'significant statements' were determined, how themes were generated, and whether a single investigator or a collaborative process was used, etc.

The discussion should include a section on the limitations of the study, for example, the families involved in the study may not be representative of the broader PWS population in Malaysia, the limitations of the interview process and observing a meal in the setting of a behavioral lab, etc.

Minor edits/comments:

Prader-Willi syndrome should be spelled out once and the abbreviation (PWS) indicated, then “PWS” should be used throughout the manuscript

Line 55 – While growth hormone improves growth and body composition, there is no evidence (known to this reviewer) that its use alters the progression of the nutritional phases – please provide evidence for this statement or revise.

Lines 57-61 Please clarify – a nasogastric tube is generally considered a type of tube feeding, but, as written, these sentences imply that it is not.

Line 74-75 – The efficacy of behavioral management interventions for features of PWS such as food stealing, overeating and temper outburst has been limited as per most reports, in contrast to the statement that “Interventions such as rewards systems…..have been effective” Additional evidence should be provided for this statement, or it should be modified to reflect the mixed findings in the literature.

Lines 124-125 – please reword “or in what takes them at distance” – - the meaning isn’t clear.

Reviewer #2: 1. Introduction needs to be re-written and to be edited suitably.

2. Methodology needs to be corrected

3. Add the reference letter for ethical approval.

4. Grammatical errors

5. Table-3 legends are confusion.

6. PLOS authors have the option to publish the peer review history of their article (what does this mean?). If published, this will include your full peer review and any attached files.

Reviewer #1: No

Reviewer #2: **Yes: **Amitesh Narayan

---

## [Author Response · Author response to Decision Letter 0]

17 Apr 2024

- Updated the statement on funder's role in this study.

- Edited the introduction to make it more concise and better flow

- More details on methods were added.

- Additional paragraph added in discussion section

- Abrreviation PWS is used throughout the manuscript.

- Edited all sentences highlighted: Line 55. 57-61, 74-75, 124-125.

- Form for ethical approval has been attached

- Table 3 and its legend has been improved.

- Conclusion has been edited

- Edited the whole paper for grammatical errors and clarity.

---

## [Decision Letter · Decision Letter 1]

5 Jun 2024

PONE-D-23-43273R1Management of food socialization for children with Prader-Willi Syndrome: An exploration study in MalaysiaPLOS ONE

Dear Dr. Wan,

Thank you for submitting your manuscript to PLOS ONE. After careful consideration, we feel that it has merit but does not fully meet PLOS ONE’s publication criteria as it currently stands. Therefore, we invite you to submit a revised version of the manuscript that addresses the points raised during the review process.

**ACADEMIC EDITOR: The article needs minor revision before further consideration according to the reviewers.**==============================

We look forward to receiving your revised manuscript.

Kind regards,

Ranjit Kumar Dehury

Academic Editor

PLOS ONE

Journal Requirements:

Additional Editor Comments:

Dear authors,

The article needs minor revision before further evaluation.

with regards,

Ranjit

Reviewers' comments:

Reviewer's Responses to Questions

**Comments to the Author**

1. If the authors have adequately addressed your comments raised in a previous round of review and you feel that this manuscript is now acceptable for publication, you may indicate that here to bypass the “Comments to the Author” section, enter your conflict of interest statement in the “Confidential to Editor” section, and submit your "Accept" recommendation.

Reviewer #2: All comments have been addressed

2. Is the manuscript technically sound, and do the data support the conclusions?

Reviewer #2: (No Response)

3. Has the statistical analysis been performed appropriately and rigorously? 

Reviewer #2: Yes

4. Have the authors made all data underlying the findings in their manuscript fully available?

Reviewer #2: Yes

5. Is the manuscript presented in an intelligible fashion and written in standard English?

Reviewer #2: Yes

6. Review Comments to the Author

Reviewer #2: Page- 16 line 338 to 340:

About 40% of the strategies employed by the families were retrieved during the observation and were supported by the family's feedback during the reflective interview; 40% were from the reflective interview, and only 23% were from the observation.

Information presented is confusing and needs attention by the authors.

Rest are small grammatical errors.

7. PLOS authors have the option to publish the peer review history of their article (what does this mean?). If published, this will include your full peer review and any attached files.

Reviewer #2: **Yes: **Amitesh Narayan

---

## [Author Response · Author response to Decision Letter 1]

13 Jun 2024

Comments Provided Response from Authors

Reviewer #2: Page- 16 line 338 to 340:

Feedback 1: About 40% of the strategies employed by the families were retrieved during the observation and were supported by the family's feedback during the reflective interview; 40% were from the reflective interview, and only 23% were from the observation.

Information presented is confusing and needs attention by the authors.

Response: The sentence has been rephrased for clarity: 

About 23.4% of the strategies employed by the families were identified during the observation, 40.43% were identified from the reflective interview, and 36.17% were identified from both observation and reflective interview.

Feedback 2: Rest are small grammatical errors.

Response: The manuscript has been edited based on the suggestions provided.

---

## [Decision Letter · Decision Letter 2]

15 Jul 2024

Management of food socialization for children with Prader-Willi Syndrome: An exploration study in Malaysia

PONE-D-23-43273R2

Dear Dr. Wan,

We’re pleased to inform you that your manuscript has been judged scientifically suitable for publication and will be formally accepted for publication once it meets all outstanding technical requirements.

Kind regards,

Ranjit Kumar Dehury

Academic Editor

PLOS ONE

Additional Editor Comments (optional):

Dear authors,

The paper is accepted.

With regards,

Ranjit

Reviewers' comments:

Reviewer's Responses to Questions

**Comments to the Author**

1. If the authors have adequately addressed your comments raised in a previous round of review and you feel that this manuscript is now acceptable for publication, you may indicate that here to bypass the “Comments to the Author” section, enter your conflict of interest statement in the “Confidential to Editor” section, and submit your "Accept" recommendation.

Reviewer #2: All comments have been addressed

2. Is the manuscript technically sound, and do the data support the conclusions?

Reviewer #2: Yes

3. Has the statistical analysis been performed appropriately and rigorously? 

Reviewer #2: Yes

4. Have the authors made all data underlying the findings in their manuscript fully available?

Reviewer #2: Yes

5. Is the manuscript presented in an intelligible fashion and written in standard English?

Reviewer #2: Yes

6. Review Comments to the Author

Reviewer #2: 1. Good findings.

2. Applicable for the families and professionals,

3. Think of broadening its findings across the Asian countries.

7. PLOS authors have the option to publish the peer review history of their article (what does this mean?). If published, this will include your full peer review and any attached files.

Reviewer #2: **Yes: **Amitesh Narayan

---

## [Editor Report · Acceptance letter]

25 Jul 2024

PONE-D-23-43273R2 

PLOS ONE

Dear Dr. Wan, 

I'm pleased to inform you that your manuscript has been deemed suitable for publication in PLOS ONE. Congratulations! Your manuscript is now being handed over to our production team.

Kind regards, 

on behalf of

Dr. Ranjit Kumar Dehury 

Academic Editor

PLOS ONE